# *Cupriavidus metallidurans* CH34 Possesses Aromatic Catabolic Versatility and Degrades Benzene in the Presence of Mercury and Cadmium

**DOI:** 10.3390/microorganisms10020484

**Published:** 2022-02-21

**Authors:** Pablo Alviz-Gazitua, Roberto E. Durán, Felipe A. Millacura, Franco Cárdenas, Luis A. Rojas, Michael Seeger

**Affiliations:** 1Laboratorio de Microbiología Molecular y Biotecnología Ambiental, Departamento de Química & Centro de Biotecnología, Universidad Técnica Federico Santa María, Avenida España 1680, Valparaíso 2390123, Chile; alviz.pablo@gmail.com (P.A.-G.); ro.duran.vargas@gmail.com (R.E.D.); f.millacura.a@gmail.com (F.A.M.); soyfranco@gmail.com (F.C.); 2Departamento de Ciencias Biológicas y Biodiversidad, Universidad de los Lagos, Osorno 5311890, Chile; 3School of Biological Sciences, University of Edinburgh, Edinburgh EH9 3JQ, UK; 4Centro Regional de Estudios en Alimentos Saludables (CREAS), Avenida Universidad 330, Curauma, Valparaíso 2373223, Chile; 5Departamento de Química, Facultad de Ciencias, Universidad Católica del Norte, Avenida Angamos 610, Antofagasta 1270709, Chile; l.rojas@ucn.cl

**Keywords:** *Cupriavidus metallidurans*, aromatic catabolism, benzene, mercury, cadmium, bacterial multicomponent monooxygenase

## Abstract

Heavy metal co-contamination in crude oil-polluted environments may inhibit microbial bioremediation of hydrocarbons. The model heavy metal-resistant bacterium *Cupriavidus metallidurans* CH34 possesses cadmium and mercury resistance, as well as genes related to the catabolism of hazardous BTEX aromatic hydrocarbons. The aims of this study were to analyze the aromatic catabolic potential of *C. metallidurans* CH34 and to determine the functionality of the predicted benzene catabolic pathway and the influence of cadmium and mercury on benzene degradation. Three chromosome-encoded bacterial multicomponent monooxygenases (BMMs) are involved in benzene catabolic pathways. Growth assessment, intermediates identification, and gene expression analysis indicate the functionality of the benzene catabolic pathway. Strain CH34 degraded benzene via phenol and 2-hydroxymuconic semialdehyde. Transcriptional analyses revealed a transition from the expression of catechol 2,3-dioxygenase (*tomB*) in the early exponential phase to catechol 1,2-dioxygenase (*catA1* and *catA2*) in the late exponential phase. The minimum inhibitory concentration to Hg (II) and Cd (II) was significantly lower in the presence of benzene, demonstrating the effect of co-contamination on bacterial growth. Notably, this study showed that *C. metallidurans* CH34 degraded benzene in the presence of Hg (II) or Cd (II).

## 1. Introduction

Benzene, toluene, ethylbenzene, and xylene isomers, commonly known as BTEX, are volatile monoaromatic hydrocarbons often present in crude oil spilled sites and polluted industrial areas [1,2]. These compounds can easily spread to distant locations from their original locations, polluting natural water bodies, groundwater, and atmospheric air [3]. BTEX are hazardous to living organisms; their toxicity, mutagenicity, and carcinogenic effects increase during bioaccumulation in animal and human tissues. Benzene is the most unsafe and toxic component of this group, causing cancer and leukemia in humans [4]. BTEX are also toxic for microorganisms, disturbing the cell membranes, disrupting the electron transport systems, and provoking oxidative stress damage [2,5].

In polluted sites, bacteria are often exposed to a complex scenario wherein co-contamination of hydrocarbons and heavy metal ions may reduce the degradation potential of bacterial strains and effectiveness of the bioremediation process [6,7,8,9,10,11,12]. Aerobic bacterial degradation of hydrocarbons may be inhibited by heavy metals such as mercury and cadmium [6,13,14,15,16]. Cadmium is a by-product of non-ferrous metal ore processing, a component of battery manufacturing, and a plastic stabilizer [17,18]. Mercury has been extensively used in gold extraction by amalgam and is released from heavy metal mining (e.g., copper, silver, and gold) and industrial activities [11,19,20,21]. BTEX biodegradation in the presence of cadmium and mercury has barely been studied. Cd (II) was found to cause inhibition of toluene, ethylbenzene, and *o*-xylene degradation by *Bacillus* sp. [22], while limited studies have assessed the role of Hg on BTEX bacterial catabolism. Cadmium and mercury are toxic for microorganisms and may affect their bioremediation capabilities, mainly through the binding to sulfhydryl groups (cysteine and methionine) and substitutive metal binding to metalloproteins [8,22,23,24,25].

Diverse *Cupriavidus* strains are attractive candidates for bioremediation under adverse conditions, due to their aromatic compound catabolic versatility and heavy metal resistance [26,27,28,29,30,31,32]. *Cupriavidus pinatubonensis* JMP134 is a model strain for the degradation of chloroaromatics, chlorophenols, halobenzoates, nitrophenols, and 2,4-D, exhibiting a wide enzymatic machinery to overcome aromatic compounds in the environment [27]. *Cupriavidus necator* NH9 is capable of degrading 3-chlorobenzoate (3-CBA) [31], while the biphenyl and chlorpyrifos degraders *Cupriavidus basilensis* KF708 and *Cupriavidus nantongensis* X1 have also recently been characterized [29,33]. The recombinant strain *Cupriavidus pinatubonensis* JMS34 has been patented for the bioremediation of polychlorobiphenyls (PCBs) [34]. On the other hand, *Cupriavidus metallidurans* strains possess resistance to a broad range of heavy metals and have been applied in mercury bioremediation trials, whereas specific strains exhibit growth on aromatic compounds [11,21,25,35,36].

*C. metallidurans* CH34 is a model heavy metal-resistant bacterium isolated from a zinc decantation tank factory sediment. Strain CH34 harbors a genome composed of four replicons: one chromosome (C1), one chromid (C2), and two large plasmids (pMOL28 and pMOL30) [35,37,38,39]. Both megaplasmids carry the genetic determinants for heavy metal resistance, including the *mer* genes involved in mercury detoxification and cadmium resistance [39,40]. The metabolically modified strain *C. metallidurans* strain MSR33 has been patented for the bioremediation of sites polluted with mercury, cadmium, and copper [41]. Furthermore, diverse catabolic gene clusters have been reported in *Cupriavidus* genomes [31,39,42,43,44], including the genomic island CMGI-2 responsible for toluene degradation in *C. metallidurans* CH34 [30]. Previous studies have predicted the CH34 catabolic potential to grow on monoaromatic compounds (i.e., benzene, toluene, *o*-xylene, phenol, vanillate, *L*-tyrosine, 4-hydroxyphenylacetate), but only its growth on toluene, benzoate, 4-hydroxybenzoate, *L*-tyrosine, vanillate, phenylacetate, and phenylpyruvate have been experimentally reported [30,36,39,44,45,46,47].

Genomic analyses of *Cupriavidus metallidurans* strain CH34 revealed three bacterial multicomponent monooxygenases (BMM), which are a rare trait and may lead to new mixed or hybrid BMM with novel substrate specificities [48]. The aims of this study were to analyze the aromatic catabolic potential of *C. metallidurans* CH34 and to determine the functionality of the predicted benzene catabolic pathway and the influence of cadmium and mercury on benzene degradation.

## 2. Materials and Methods

### 2.1. Chemicals

Benzene, toluene, phenol, 3-hydroxyphenylacetate (3-HPA), 4-hydroxyphenylacetate (4-HPA), HgCl_2_, and CdCl_2_ of analytical grade were obtained from Merck (Darmstadt, Germany). Benzoate (BA), 3-chlorobenzoate (3-CBA), 4-chlorobenzoate (4-CBA), 3,5-dichlorobenzoate (3,5-CBA), 2-hydroxybenzoate (salicylate), 3-hydroxybenzoate (3-HBA), 4-hydroxybenzoate (4-HBA), 4-isopropylbenzoate (*p*-cumate), vanillin, cinnamate, *L*-phenylalanine, phenylacetate, anthranilate, 4-aminobenzoate (*p*ABA), benzamide, nitrobenzene, catechol, gallate, hydroxyquinol (HQ), *p*-cymene (>98% purity), ethylbenzene, *o*-xylene, *m*-xylene, *p*-xylene, and sodium succinate were obtained from Sigma-Aldrich (St. Louis, MO, USA).

### 2.2. Bacterial Strains and Culture Conditions

*C. metallidurans* CH34 [37] and *Pseudomonas putida* F1 [49] were cultivated in low-phosphate Tris-buffered mineral salts (LPTMS) medium at 30 °C. The LPTMS medium contained (per 1 L) 6.06 g Tris, 4.68 g NaCl, 1.07 g NH_4_Cl, 1.49 g KCl, 0.43 g Na_2_SO_4_, 0.2 g MgCl_2_ × 6H_2_O, 0.03 g CaCl_2_ × H_2_O, 0.23 g Na_2_HPO_4_ × 12H_2_O, 0.005 g Fe (III)/(NH4) citrate, and 1 mL of trace element solution SL7 of Biebl and Pfennig [35,37]. Succinate (6.2 mM), benzene (5 mM), or other aromatic compounds (2 mM) were used as sole carbon and energy sources, provided directly in the liquid phase unless otherwise indicated. To avoid BTEX and *p*-cymene volatilization, we closed tubes or flasks using PTFE/silicone-screw caps (DURAN Group GmbH, Wertheim, Germany).

### 2.3. Growth of C. metallidurans CH34 on Benzene and Other Aromatic Compounds

For benzene and aromatic compound growth experiments, CH34 cells grown in LPTMS medium on succinate (overnight culture), washed twice with NaCl 0.85% *p*/*v* and suspended in saline solution NaCl 0.85% *p*/*v*, were used as inoculum. Growth was followed by measuring the turbidity at 600 nm.

Strain CH34 was cultured in LPTMS medium using benzene (5 mM) as the sole carbon and energy source in 250 mL flasks with PTFE/silicone-screw caps at 150 rpm. These cultures were carried out after an adaptation protocol consisting in successive passages in LPTMS medium using 0.5 mM and 2.0 mM benzene using succinate-grown cells as initial inoculum (1% *v/v*, initial turbidity_600nm_ ≈ 0.006). Strain CH34 growth on benzene was determined by measuring turbidity at 600 nm. Values were calculated as the mean ± SD of the results of at least three independent experiments.

Growth on other aromatic compounds (toluene, ethylbenzene, xylenes, *p*-cymene, phenanthrene, anthracene, nitrobenzene, BA, 3-CBA, 4-CBA, 3,5-CBA, salicylate, 3-HBA, 4-HBA, *p*ABA, anthranilate, *p*-cumate, PA, 3-HPA, 4-HPA, vanillin, cinnamate, gallate, *m*-toluate, and HQ) was assessed in fresh LPTMS using each compound (2 mM) as sole carbon and energy source and succinate-grown cells as inoculum (1% *v/v*, initial turbidity_600nm_ ≈ 0.006). Growth was assessed every 24 h for 7 days by measuring turbidity_600nm_.

During growth on some aromatics, bacterial aggregation was identified by naked-eye observation. For BTEX compounds and *p*-cymene, growth by CH34 cells was also assessed in LPTMS agar plates containing sterile-filter papers soaked with each compound placed on the inside cover plate to supply the hydrocarbon via vapor-phase. Values were calculated as the mean ± SD of the results of at least three independent experiments.

### 2.4. Aromatic Catabolism Reconstruction of C. metallidurans Strain CH34

*C. metallidurans* CH34 genome was retrieved from NCBI (Accession Nº GCA_000196015.1). A curated dataset of catabolic enzymes (ring-hydroxylating oxygenases, multicomponent monooxygenases, extradiol, and intradiol dioxygenases) involved in aromatic compound catabolism reported in UniProtKB-Swissprot, EAWAG, KEGG, and scientific literature were analyzed in CH34 genome using a best-bidirecctional hit (BBH) approach with the BLASTP and TBLASTN tools of BLAST 2.7.1. software [50]. An identity ≥ 30%, coverage ≥ 70%, and e-value cut-off ≤ 1 × 10^−10^ were used as a filter for gene identification. The genomic context of each gene identified was analyzed to identify a complete aromatic catabolic pathway. Missing enzymes of an incomplete pathway were manually searched on the CH34 genome using the same BBH approach and filter parameters.

### 2.5. Comparative and Phylogenetic Analysis of Bacterial Multicomponent Monooxygenase (BMM) Gene Clusters

The organization of BMM gene clusters involved in the BTEX degradation by *C. metallidurans* CH34 was compared with reported BTEX-degrading bacteria [27,51,52,53,54,55,56]. Nucleotide sequences of BMM gene clusters (*dmp*, *phh*, *phl*, *phy*, *aph*, *tbm*, *tbc*, *tbu*, *tom*, *tmo*, and *tbc* gene clusters) were retrieved from the GenBank database (Appendix A). In some cases, the information was updated with genomic data. BMM gene clusters were aligned and organized according to the phylogenetic relationship of the oxygenase large subunit and the most divergent sequence (*phyC*, *dmpN*, *tbmD*, *phlN*, *aphN*, *tomA3*, *tomE*, *tbuA2*, *tbc2E*). Amino acid sequences were aligned using MAFFT version 7.407 [57] and manually trimmed using Aliview version 1.24 [58]. A Bayesian inference phylogenetic mid-rooted tree was obtained using MrBayes 3.2.6 (250,000 generations, two chains each run; sampling every 1000 generations) [59]. Visualization and editing of phylogenetic trees were performed using the FigTree v.1.4.2 software (http://tree.bio.ed.ac.uk/software/figtree/, accessed on 21 March 2020). Bootstrap values (percentage) above 50% are shown for each branch point.

### 2.6. RNA Isolation and Gene Expression Analysis

Total RNA was isolated from *C. metallidurans* CH34 cells grown until early exponential phase (turbidity_600nm_ = 0.2–0.3) and late exponential phase (turbidity_600nm_ = 0.5–0.6) on benzene (5 mM) or succinate (6.2 mM) using the RNeasy mini kit (Qiagen, Hilden, Germany) according to the manufacturer’s recommendations. TURBO DNAfree set (LifeTechnologies, Carlsbad, CA, USA) was used to degrade residual DNA. A final qPCR test with the *gyrB* gene primers designed by Primer 3.0 (Appendix A) was performed to confirm genomic DNA absence. The RNA concentration was quantified using a Qubit fluorometer (Invitrogen, Carlsbad, CA, USA) and a Nanodrop spectrophotometer (Thermo Scientific, Carlsbad, CA, USA). RNA integrity was observed by agarose (1%) gel electrophoresis.

Reverse transcription was carried out using 200 ng of RNA and a High Capacity cDNA Reverse Transcription Kit (Applied Biosystems, Foster City, CA, USA). The Minimum Information for Publication of Quantitative Real-Time PCR Experiments (MIQE) guideline was used as standard protocol [60]. Real-time PCR was performed using 20 ng of cDNA on a StepOne Real-Time PCR System (Applied Biosystems, Foster City, CA, USA), using Maxima SYBR Green/ROX qPCR Master Mix (Thermo Scientific, Carlsbad, CA, USA) and 0.3 µM of each primer. cDNA was initially denatured at 95 °C for 5 min. A 40 cycle amplification and quantification protocol (95 °C for 15 s, 55 °C for 15 s, and 60 °C for 15 s) with a single fluorescence measurement per cycle followed by a melting-curve program (95 °C for 15 s, 25 °C for 1 s, 50 °C for 15 s, and 95 °C for 1 s) were used according to the manufacturer’s recommendations. PCR melting curves confirmed the amplification of a single product for each primer pair. Primers yielded products between 200 and 250 bp. The *gyrB* gene (*RMET_RS00030*) was amplified as a reference gene, yielding an amplicon of 233 bp. A standard curve was made in triplicate using serial dilutions (10-fold) for each amplicon in a linear range (10 ng–0.1 pg) of genomic DNA. qPCR efficiencies were calculated from the slopes of the log-linear portion of calibration curves by using the equation E = 10(1/slope). The reference *gyrB* gene was stably expressed according to the algorithms of BestKeeper [61]. Relative gene expression ratios were determined as outlined by Pfaffl (2001) [62], thereby normalizing gene expression levels of CH34 cells grown on benzene versus cells grown on succinate.

### 2.7. Determination of Mercury and Cadmium Minimum Inhibitory Concentrations

The resistance to mercury and cadmium for strains CH34 and F1 was determined by the value of minimum inhibitory concentration (MIC) of metals that prevent growth [35,37]. In order to assess MIC of *C. metallidurans* CH34 and *P. putida* F1 using succinate as sole carbon and energy source, we grew the cells until the late exponential phase in LPTMS medium using succinate (6.2 mM) as a carbon source and inoculated them in fresh LPTMS medium at the same succinate concentration at initial turbidity_600 nm_ = 0.06. On the other hand, for MIC determination using benzene (5 mM) as the sole carbon source, the cells were subjected to a previous adaptation protocol to induce the expression of the catabolic machinery prior to heavy metal exposure. This adaptation consisted of benzene exposure to succinate-grown cells, as previously described, inoculating the cells in fresh LPTMS with benzene (1 mM) and 12 h incubation under the same culture conditions. Benzene-adapted cells were subsequently washed twice using NaCl (0.85% *p*/*v*) and Tween20 (0.05% *v/v*) in order to eliminate remaining benzene and inoculated in fresh LPTMS medium using benzene (5 mM) as the sole carbon and energy source to reach an initial turbidity_600 nm_ = 0.006 in glass tubes closed using PTFE/silicone-screw caps. Succinate and benzene-grown cells were exposed to increasing concentrations of Hg (II) or Cd (II) from stock solutions of HgCl_2_ and CdCl_2_. Data were collected after 48 h at 30 °C. Values were determined using at least three independent biological replicates.

### 2.8. Benzene Degradation Assays

Benzene degradation assays were performed using borosilicate glass tubes (25 mL) with PTFE/silicone-screw caps and arranged in horizontal position. Glass tubes were equilibrated with 1.25 mL of resting cell buffer (Tris-Cl 50 mM (pH 7.0), Tween20 (0.05% *v/v*)) supplemented with benzene (10 mM) for 12 h under culture conditions at 30 °C. Tween20 (0.05% *v/v*) was added to increase bioavailability in buffer solution and mitigate volatilization. *C. metallidurans* CH34 and *P. putida* F1 cells were subjected to the benzene adaptation protocol described in Section 2.7. Washed cells were concentrated to obtain a cell suspension at turbidity_600nm_ = 10 in the resting cell buffer. Aliquots of 1.25 mL of cell suspensions were added to the benzene-equilibrated glass tubes, resulting in a 2.5 mL resting cell suspension at turbidity_600nm_ = 5.0 and benzene (5 mM). Final concentrations of Hg (II) and Cd (II) were obtained after the addition of aliquots from HgCl_2_ and CdCl_2_ 100 × stock solutions. Samples (100 µL) were taken at 3 h intervals during 24 h of incubation at culture conditions. Cells were removed by centrifugation (19,000× *g* for 1 min) for subsequent analyses of benzene and their metabolites by high-performance liquid chromatography coupled to a diode array detector (HPLC-DAD) and spectrophotometry. A boiled cell suspension was used as a dead cell control to determine abiotic loss of benzene. Cells were killed by boiling for 10 min in a water bath. Resting cells assays were performed in triplicate, and values were calculated as the mean ± SD of the results of at least three independent experiments.

HPLC-DAD analysis was carried out using a Jasco HPLC model LC-2000 equipped with a diode array detector (DAD) Jasco model MD-2015 using a RP 18e/Chromolith column of 100–4.6 mm and a pore size of 13 nm (Merck, Darmstadt, Germany). The solvents used for sample elution were 0.1% formic acid in water (A) and 100% acetonitrile (B). The flow rate was 1.0 mL/min, and the elution profile was 70% A/30% B for 4 min, then changed linearly to 0% A/100% B over a 1 min period and kept at this ratio for 3 min and finally changed linearly to 30% A/70% B over 1 min and kept at this ratio for 2 min. Benzene, phenol, and catechol concentrations were monitored using calibration curves with authentic standards. The injection volume was 10 µL. Benzene, phenol, and catechol determinations were performed at 200, 270, and 276 nm wavelengths, respectively.

2-Hydroxymuconic semialdehyde (2-HMS) content was quantified by measuring absorbance at 375 nm by spectrophotometry using an Infinite 200 TECAN microplate reader (Tecan Group Ltd., Männedorf, Switzerland). The quantification was based on the molar absorption coefficient 3.3 × 10^4^ of 2-HMS [63].

## 3. Results

### 3.1. Genome-Based Reconstruction of the Aromatic Catabolism in Strain CH34

The aromatic catabolism of *C. metallidurans* CH34 was reconstructed on the basis of its genome and using the genomic information of aromatic-degrading bacteria. CH34 genome carries 112 genes encoding 10 central and 19 peripheral pathways/reactions (Figure 1). All the genes identified are located on C1 and C2, whereas genes for aromatic catabolism were not observed on plamids pMOL28 and pMOL30. Ten aromatic central catabolic pathways were encoded by 65 genes. The aromatic central catabolic pathways include the catechol and protocatechuate branches of the β-ketoadipate pathway (*cat* and *pca* genes), *meta*-cleavage catechol (*tom* genes), gallate (*gal* genes), HQ (*pnp* genes), homogentisate (*hmg* genes), 3-hydroxyanthranilate (*onb* genes), 2-aminobenzoyl-CoA (*abm* genes), phenylacetyl-CoA (*paa* genes), and benzoyl-CoA (*box* genes) (Appendix A). The genes of aromatic peripheral catabolic pathways/reactions comprised chlorobenzoate (*ben* genes), benzoate (*ben* and *bcl* genes), 4-HBA (*pob* genes), phenylalanine (*phh* genes), tyrosine (*tyrB*), 2-hydroxyphenylacetate (2-HPA; *ohpA*), anthranilate (*abmG*), 3-hydroxy-*L*-kynurenine (*kynU*), tryptophan (*kyn* genes), 4-hydroxyphenylpyruvate (*hppD*), phenylacetate (*paaK*), coniferyl alcohol (*calA*), coniferyl aldehyde (*calB*), ferulic acid (*fcs*), vanillin (*vdh*), vanillate and isovanillic acid (*iva* genes), phenol (*phy* genes), and BTEX compounds (*phy*, *tmo*, and *tom* genes) (Figure 1, Appendix A).

The genomic islands (GIs) CMGI-1 (*RMET_RS11485*–*RMET_RS12070*), CMGI-2 *RMET_RS06240*–*RMET_RS06810*), and CMGI-E (*RMET_RS27890*–*RMET_RS28445*) are the only GIs encoding genes for aromatic peripheral and central catabolic pathways in the *C. metallidurans* CH34 genome. The chromosome (C1) of strain CH34 contains the CMGI-1 locus, which includes a crotonase/enoyl-CoA hydratase-encoding gene (*ech*) probably involved in the catabolic pathway of hydroxycinnamic acids to the central intermediate protocatechuate (Figure 1). C1 also harbors CMGI-2, which is related to benzene/toluene catabolism [30,39,44]. CMGI-2 locus is a gene cluster of 24,547 bp encompassed from *RMET_RS06580* to *RMET_RS06715* (Appendix A), encoding two bacterial multicomponent monooxygenases (BMMs, *tomA0A1A2A3A4A5* and *tmoABCDEF*), one catechol 2,3-dioxygenase (*tomB*), one transcriptional regulator (XylR/NtrC-type), a membrane transport protein (TbuX/FadL-type), and seven additional enzymes involved in the central metabolism of catechols and methylcatechols (Figure 2). The chromid (C2) of strain CH34 possesses only one genomic island (CMGI-E) related to aromatic compound degradation, encoding the genes for an isovanillate/vanillate *O*-demethylase (*ivaAB*) and 4-hydroxybenzoate-3-monooxygenase (*pobA2*) (Figure 1, Appendix A). A total of 27 of the 112 genes identified for aromatic catabolism in *C. metallidurans* CH34 are present in GIs (Appendix A).

A third BMM is located in the CH34 genome, encoded in a gene cluster of 7587 bp from *RMET_RS08920* to *RMET_RS08955* between the CMGI-11 and CMGI-6 (Appendix A) [40]. This genomic locus comprises a phenol hydroxylase (*phyZABCDE*), a catechol 1,2-dioxygenase (*catA1*), and a XylR/NtrC-type transcriptional regulator (*poxR*) [39]. A phylogenetic analysis of the amino acid sequence of the most divergent oxygenase component from *C. metallidurans* CH34 and other Proteobacteria allowed for the classification of the three BMMs. PhyZABCDE and TomA0A1A2A3A4A5 were classified into group I, whereas TmoABCDEF belonged to the second BMM group (Figure 2). Group I of proteobacterial BMMs was highly represented by phenol hydroxylases (PHs), including two clades completely composed of PHs (green and orange), and wherein PhyC from the *phyZACBDE* gene cluster was located (Figure 2A). PHs are responsible for phenol and benzene catabolism in other Proteobacteria [50]. The closest sequence to PhyC from strain CH34 is PhlN1, which is part of the *phl* gene cluster of *C. pinatubonensis* JMP134, another gene cluster that also encodes a catechol 1,2-dioxygenase (*catA*) downstream of the operon. A distinct clade inside group I (blue, Figure 2A) formed by the PHs toluene-benzene 2-monooxygenases (TB2MOs) and toluene 2-monooxygenases (T2MOs) contains TmoA3 from strain CH34 (first BMM from the CMGI-2 aromatic gene cluster). The TmoA3 clade is composed of two T2MOs belonging to *Burkholderia vietnamiensis* G4 and *C. metallidurans* CH34, possessing high synteny and identity between both T2MO gene clusters and the downstream *meta*-cleavage catechol pathway (*tomXBYCEFGHI*; Appendix A). Interestingly, the *B. vietnamensis* G4 cluster is located on the plasmid pBVIE04 and carries an ISBmu2-like-insertion sequence (Figure 2A). The last CH34 BMM is also part of the first 24 kb gene cluster from CMGI-2 and was classified as part of the group II BMMs (Figure 2B). Group II of BMM was composed of broad-substrate monooxygenases such as benzene monooxygenases (BMO), toluene-2-monooxygenases (T2MO), toluene-3-monooxygenases (T3MO), toluene-4-monooxygenases (T4MO), and toluene/*o*-xylene monooxygenases (ToMO). For example, ToMO oxidizes benzene; toluene; *o*-, *m*- and *p*-xylene; ethylbenzene; 2,3- and 3,4-dimethylphenol; cresols; naphthalene; TCE; and chloroform [54]. The TmoE γ-hydroxylase subunit of strain CH34 does not cluster with other known group II BMMs (Figure 2B).

The *tmoX* (*RMET_RS06590*) gene, part of the CMGI-2, encodes a TbuX/FadL-type outer membrane transport protein that possesses a 43% identity with the transporter TmoX of *Pseudomonas mendocina* KR1, which is part of the toluene X superfamily of mono-aromatic outer membrane transport proteins [64].

### 3.2. C. metallidurans CH34 Growth on Aromatic Compounds

In LPTMS minimal medium, *C. metallidurans* CH34 was able to grow on benzene, toluene, *p*-cymene, BA, 3-CBA, 2-hydroxybenzoate (salicylate), 3-HBA, 4-HBA, *p*ABA, PA, 3-HPA, 4-HPA, vanillin, and cinnamate as sole carbon and energy sources (Table 1). Strain CH34 did not exhibit an increase in turbidity_600nm_ in specific recalcitrant aromatic compounds, such as 3,5-CBA, ethylbenzene, and xylenes, but growth as bacterial aggregation was observed. In contrast, strain CH34 was not able to grow on gallate, phenanthrene, anthracene, nitrobenzene, anthranilate, 4-CBA, *m*-toluate, *p*-cumate, and HQ as sole carbon and energy sources (Table 1). Strain CH34 tolerated exposure to benzene at concentrations up to its saturation point in water (20 mM).

### 3.3. Benzene Catabolism by C. metallidurans CH34

The functionality of the benzene catabolic pathway of *C. metallidurans* CH34 was evaluated. Benzene was selected as a model BTEX compound due to its environmental relevance, high toxicity, and presence in hydrocarbon-polluted sites. During *C. metallidurans* CH34 growth on benzene, at early exponential phase (22 h), a yellow color formation was observed. The absorbance spectrum of the yellow-colored culture by spectrophotometry indicated the formation of 2-hydroxymuconic semialdehyde (2-HMS) during the *meta*-cleavage pathway by the catechol 2,3-dioxygenase (C23O) TomB (Figure 3B).

To identify other metabolic intermediates of the benzene catabolism, we analyzed culture supernatants of CH34 cells grown on benzene (5 mM) by HPLC-DAD. Benzene concentration decreased during the incubation, while the formation of phenol was detected after 15–26 h (Figure 3B). However, catechol was not detected by HPLC-DAD.

### 3.4. Transcriptional Analysis during Benzene Degradation

The expression of genes encoding the T2MO (*tomA3*), TMO (*tmoA*), PH (*phyC*), catechol 1,2-dioxygenase (C12O; *catA1* and *catA2*), C23O (*tomB*), semialdehyde-2-hydroxymuconate dehydrogenase (*tomC*), and semialdehyde-2-hydroxymuconate hydrolase (*tomD*) were quantified by RT-qPCR analysis. Transcriptional analysis of CH34 cells grown on benzene as the sole carbon source showed a simultaneous expression of genes encoding monooxygenases in both early and late exponential phase (Figure 4). The *tomB*, *catA1* and *catA2* genes displayed differential transcriptional expression. The *tomB* gene encoding C23O was induced at the early and late exponential phases, whereas the *catA1* and *catA2* genes showed a higher induction at the late exponential phase. The *tomC* and *tomD* genes were expressed at the early and late exponential phases, suggesting that the hydrolytic and the 4-oxalocrotonate branches of the *meta*-cleavage pathways are active. In addition, the transcription of the sigma-38 factor *rpoS* gene, as well as the LysR-type (*catM* and *benM*) and the XylR/NtrC-type (*tomR* and *poxR*) transcriptional regulators, were studied. The *tomR* gene encoding a XylR/NtrC-type transcriptional regulator was expressed at early and late exponential phase. Conversely, the sigma-38 factor *rpoS* gene, and the *benM*, *catM*, and *poxR* genes encoding for transcriptional regulators were expressed only during the late exponential phase (Figure 4). On the basis of these results, we propose that benzene is degraded in *C. metallidurans* CH34 by an upper pathway that involves successive monooxygenation reactions catalyzed by the BMMs: PH, T2MO, and TMO, producing phenol and catechol as metabolic intermediates (Figure 5). The central pathway starts with a *meta*- or *ortho*-cleavage by catechol dioxygenases TomB in order to form 2-HMS, or by CatA1 and CatA2 in order to form *cis*-*cis* muconate, respectively. Transcriptional data and the metabolite formation suggest a predominant *meta*-cleavage pathway via C23O, while an induction of the *catA1* and *catA2* genes was observed at a late exponential phase. The induction of the *tomC* and *tomD* genes at early exponential phase suggest that both branches of the *meta*-cleavage pathway are active during benzene degradation (Figure 5).

### 3.5. C. metallidurans CH34 and P. putida F1 growth on Benzene in the Presence of Mercury and Cadmium

To evaluate the effects of mercury and cadmium during growth on benzene, we determined the MICs of Hg (II) and Cd (II) in *C. metallidurans* CH34, and the model benzene-degrader *P. putida* F1. *P. putida* F1 was incorporated as a model proteobacterium for aerobic benzene degradation [52,65].

The MICs of Hg (II) and Cd (II) were significantly lower when the strains were previously grown on benzene (5 mM) compared to succinate-grown cells (Table 2). Strain CH34 showed higher MICs for Hg (II) than strain F1, independently of the carbon source. For strain F1, the impact of benzene growth on MIC was higher in the presence of Cd (II) than in presence of Hg (II), reaching only 10% of the MIC observed during growth on succinate (Table 2).

### 3.6. The Effects of Mercury or Cadmium on Benzene Degradation by C. metallidurans CH34 and P. putida F1

The effects of mercury (32.5 µM) or cadmium (200 µM) on benzene degradation by *C. metallidurans* CH34 was studied and compared with the degradation by benzene-degrader *P. putida* F1. The metal ion concentrations were selected after preliminary experiments to determine which concentration caused an inhibitory effect in bacterial catabolism. Benzene degradation is achieved in strain CH34 via BMMs, while strain F1 activates the aromatic ring using a benzene/toluene dioxygenase [66]. The benzene, 2-HMS, phenol, and catechol concentrations were monitored in resting cell assays (Figure 6, Figure 7, and Appendix A). The resting cell assay considered an initial concentration of 5 mM of benzene and a cellular turbidity_600nm_ = 5.0.

In the absence of heavy metals, *C. metallidurans* CH34 showed a lower benzene degradation rate in comparison with *P. putida* F1, which showed no residual benzene with the concomitant highest 2-HMS concentration after 3 h (Figure 6). A decrease in benzene was observed for dead cells for both strains, indicating abiotic benzene loss from the aqueous phase, probably by volatilization and biosorption. For the resting cell assay of benzene by strain CH34, phenol was detected at 3 h (Appendix A), while 2-HMS was detected at 6 h (Figure 6A and Figure 7A), supporting the proposed benzene degradation pathway in *C. metallidurans* CH34.

The presence of mercury or cadmium had no detrimental effects on benzene degradation by *C. metallidurans* CH34 (Figure 6 and Figure 7, Appendix A). Of note, despite a decay in phenol production by both heavy metals (Appendix A), mercury exposure slightly stimulated the benzene degradation, supported by an increase of 2-HMS production at 6 h incubation (Figure 6A, Appendix A). After longer incubation, CH34 2-HMS production rate was lower during Hg (II) exposure compared to the control condition. In contrast, cadmium slightly inhibited CH34 production of 2-HMS at shorter incubations (6 and 9 h), but not after 12 h (Figure 7A, Appendix A). In *P. putida* F1, the presence of mercury strongly inhibited benzene degradation, showing almost negligible 2-HMS production (Figure 6B, Appendix A). Cadmium showed no effect on strain F1 benzene oxidation; however, a slight inhibition in 2-HMS degradation was observed (Figure 7B, Appendix A).

## 4. Discussion

### 4.1. Aromatic Catabolic Reconstruction of C. metallidurans CH34

In this study, we reported a wide variety of aromatic compounds that can be used as carbon source by *C. metallidurans* CH34, and the aromatic catabolic pathways including the routes to degrade BTEX, chlorobenzoates, and lignin-derived compounds were inferred from genomic studies. We also reported new metabolic insights to describe the benzene catabolic pathways observing intermediates at different growth stages and by functional expression of key genes. Finally, we studied the effects of the presence of heavy metals on benzene degradation since the capabilities of strain CH34 to degrade aromatic compounds such as benzene in the presence or absence of toxic heavy metals have been scarcely reported in the wild-type strain CH34 [44]. Springael et al. transferred the plasmid pSS50, which is involved in polychlorobiphenyl (PCB) degradation, into strain CH34 to study the effect of nickel and zinc ions on PCB degradation by the transconjugant strain AE707, overlooking the innate catabolic potential of strain CH34 [26,45]. Aromatic degradation by *Cupriavidus* strains has been described, but only three genomes have been sequenced and analyzed to prospect their overall aromatic degradation capabilities: the chloroaromatic degrader *C. pinatubonensis* JMP134, the lignin degrader *C. basilensis* B-8, and the 3-CBA degrader *C. necator* NH9 [27,67,68]. Of note are the harboring of C1 and C2 genes by *C. metallidurans* CH34, which confers a similar catabolic versatility to strain JMP134, a well-known aromatic degrader model [69], including multiple central pathways for aromatic degradation (the protocatechuate and the catechol β-ketoadipate pathways; the *meta*-cleavage catechol pathway; the HQ pathway; the homogentisate pathway; the 3-hydroxyanthranilate pathway; and the hybrid aerobic pathways 2-aminobenzoyl-CoA, phenylacetyl-CoA, and benzoyl-CoA), three BMMs, and genes encoding benzoate and lignin-derived degradation enzymes. Morever, these results showed the ability of strain CH34 to grow on 3-CBA and 3,5-CBA despite the absence of canonical chlorocatechol pathways (*tfd* and *clc* genes). Following these results, previous studies reported that C12O of strain CH34 possesses a unique spectrum to cleave substituted-catechols in *ortho* position, including tetrachlorocatechol, 4-fluorocatechol, 4-methylcatechol, and 3-methylcatechol [70,71]. Even though specific pathways for chloro-substituted aromatics were not identified in the CH34 genome, the benzoate 1,2-dioxygenase BenABCD may perform the dioxygenase activation with a lower affinity to convert 3-CBA and 3,5-CBA into chlorocatechols, which has been reported during degradation of 3-CBA by *C. pinatubonensis* JMP134 [72]. Transcriptome profiling of *C. necator* NH9 cells grown on 3-CBA compared to benzoate supports the hypothesis that 3-CBA and benzoate are degraded by a common pathway [68].

In the present study, we also observed that *C. metallidurans* CH34 grew on benzene, toluene, ethylbenzene, and xylenes. In addition, CH34 cells are also able to grow on *p*-cymene, BA, 3-CBA, 3,5-CBA, salicylate, 3-HBA, 4-HBA, *p*ABA, PA, 3-HPA, 4-HPA, vanillin, and cinnamate. These results support the corresponding catabolic pathways identified by genomic analyses (Table 1). Although strain CH34 grew on salicylate, *p*-cymene, 3-HBA, 3-HPA, 4-HPA, benzamide, *p*ABA, and cinnamate, ortholog genes for their previously described catabolic pathways were not found in the CH34 genome, suggesting the presence of still undescribed catabolic pathways in the *Cupriavidus* genus. The absence of degradative canonical pathways of phenylpropanoids (cinnamate, phenylpropionates) and 4-HPA in *Cupriavidus* strains B-8, NH9, and JMP134, despite their growth on these compounds [27,31,67], indicate that members of the *Cupriavidus* genus may degrade them via novel catabolic pathways. Cytochrome P450 OhpA is a self-sufficient cytochrome involved in the transformation of 2-HPA into homogentisate, which was described recently in *C. pinatubonensis* JMP134 [73], being also present in the genome of strain CH34 (*RMET_RS25305*), evidencing the need for further characterization of novel enzymatic activities in the *Cupriavidus* genus.

### 4.2. BMMs in C. metallidurans CH34

The presence of more than one BMM observed in *C. metallidurans* strain CH34 is rare in bacteria, with its co-expression potentially leading to the formation of a complex modularity, generating new substrate specificity and providing optimized metabolic capabilities [40,48,74]. The first BMM of strain CH34 was constituted by T2MO (encoded by the *tomA012345* gene cluster), which possesses a high similarity in amino acid sequence and gene organization to the *B. vietnamiensis* G4 T2MO subunits (Figure 2 and Appendix A), suggesting a regiospecific hydroxylation of toluene into *o*-cresol and, subsequently, an oxidation into 3-methylcatechol [75,76,77]. In addition, T2MO catalyzes the oxidation of dichloroethylenes, chloroform, 1,4-dioxane, aliphatic ethers, and diethyl sulphide [76,78] and enables the formation of epoxides from a variety of alkene substrates [79]. The second CH34 BMM is the TMO encoded by the *tmoABCDEF* gene cluster, which belongs to the group II BMMs. The products of these genes are similar to the corresponding subunits of the BMMs TbcABCDEF (*C. pinatubonensis* JMP134 and *B. cepacia* JS150) and TmoABCEDF (*P. mendocina* KR1) (Figure 2). All these BMMs have different substrate specificities, enabling the oxidation of toluene and other BTEX compounds [54,80]. For example, T4MO of strain KR1 oxidizes toluene into 4-methylcatechol and catalyzes the formation of epoxides [79], as well as the oxidation of phenols and methylphenols into catechol [75]. An outstanding broad substrate group II BMM is the toluene *o*-xylene monooxygenase (ToMO) of *P. stutzeri* OX1, which oxidizes *o*-xylene, *m*-xylene, *p*-xylene, toluene, benzene, ethylbenzene, styrene, naphthalene, and tetrachloroethylene [74]. The characterization of the hydroxylation capabilities of CH34 TMO should be performed due to their potential broad substrate specificity and the classification of the CH34 TmoE subunit as a singleton in the phylogenetic analysis of γ-hydroxylase subunit of the group II BMMs. *C. metallidurans* CH34 genome encodes a third BMM, a PH that is encoded by the *phyZABCDE* gene cluster. This enzyme is also present in *C. pinatubonensis* JMP134 and transforms benzene into phenol, and then into catechol (Figure 5). In accordance, the *phyZABCDE* gene cluster is located upstream of the *catA1* gene, which encodes a catechol 1,2-dioxygenase (C12O) (Figure 2, Appendix A) [39]. The *phy-catA1* gene cluster might be controlled by PoxR, a phenol-sensitive XylR/NtrC-like transcriptional activator (Appendix A). Furthermore, an additional C12O is encoded on the C2 (*RMET_RS25045*) and belongs to the benzoate catabolic *ben-cat* gene cluster [28]. Multiple BMMs were encountered in other *Burkholderiales* degraders (*B. cenocepacia* JS150 and *C. pinatubonensis* JMP134), which could account for their exceptionally versatile catabolic capabilities [28].

### 4.3. Functionality of CH34 BMMs Associated with the Degradation of Benzene

In this study, the functionality of CH34 BMMs associated with the degradation of benzene was reported for the first time. Transcriptional analyses and kinetics of the metabolic intermediates indicated that CH34 BMM activities depend on the growth phase. Phenol accumulation during CH34 growth on benzene was observed (Figure 3B). The results of this study suggest that one or more BMMs could be activated simultaneously during the process, thereby indicating the occurrence of a complex degrading module to generate a catabolic strategy. Interestingly, activation of diverse BMMs has been proposed in *C. pinatubonensis* JMP134, *R. pickettii* PKO1, and *B. cenocepacia* JS150 [48,54,80]. RT-qPCR revealed the successive use of two transcription profiles according to the growth phase. At early exponential phase, two BMM-encoding gene clusters (*tomA012345* and *tmoABCDEF*), and *tomB*, *tomC*, and *tomD* genes forming the C23O pathway that all belong to the the CMGI-2 catabolic gene cluster (Figure 4) were induced. In contrast, at late exponential phase, the *phyZABCDE* gene cluster encoding the PH, and the *catA1* and *catA2* genes that compose the C12O pathway were upregulated. This transcription profile associated with the late exponential growth phase during benzene catabolism is in agreement with the downstream location of a *rpoS* promoter and the reported inhibition of C12O activity in the presence of phenol [54]. Therefore, these results suggest an induction of the C12O pathway only at late growth phase on benzene at ultimately lower phenol concentrations (Figure 7). Transcription analyses and metabolic intermediate detection suggest that *meta*-cleavage of catechol is catalyzed by a C23O (TomB) at early stages of growth, whereas *ortho*-cleavage, catalyzed by a C12O (CatA1, CatA2) at late stages of growth, is driven by starvation and the general stress transcriptional factor RpoS. Although versatile predominance between the C12O or C23O degradation pathways has been previously reported as dependent of substrate and concentration [81,82], more analyses should be performed to determine to which extent each CH34 catabolic pathway is involved in benzene degradation.

### 4.4. Effect of Benzene on Bacterial Cadmium MICs

Cadmium MICs of *C. metallidurans* CH34 and *P. putida* F1 grown on benzene were significantly lower compared to MICs of succinate-growing bacteria. However, cadmium showed a higher impact on strain F1. These results are in agreement with the heavy metal resistance niche specialization of the CH34 strain, reflexed on its horizontal acquisition and duplication of genes that encode cadmium efflux systems [39,83,84], some of which are represented as homologous genes in the genome of strain F1, although to a significant lower extent [85].

Previously, we have reported in strain CH34 that Cd (II) induces genes encoding the cadmium efflux system CadA and the mercury reductase MerA that are involved in heavy metal response [41]. In addition, cadmium decreases the second messenger c-di-GMP, induces a metal regulated phosphodiesterase, and inhibits biofilm formation [41]. Aromatic hydrocarbons may increase membrane fluidity, causing non-specific permeabilization and membrane protein disruption [86,87], which may affect the cation uptake and explain the reduced MIC values. The occurrence of futile cycles during cadmium detoxification, but not in mercury detoxification systems, has been reported due to the volatile nature of the mercury detoxification product [88]. Therefore, a faulty efflux system for Cd (II) may originate from benzene exposure in both CH34 and F1 strains. Cd (II) induces oxidative stress in bacteria [88]. An increase in reactive oxygen species by cadmium can lead to protein carbonylation, lipoperoxidation, and oxidative damage to DNA, which may act in synergy with benzene-induced membrane disruption, strongly reducing cell viability [89].

### 4.5. Effects of Mercury and Cadmium on Benzene Degradation

In the present study, the effects of mercury and cadmium on benzene degradation by *C. metallidurans* CH34 and the model benzene/toluene-degrader *P. putida* F1 were studied and compared. In the absence of heavy metals, *P. putida* strain F1 showed a higher benzene degradation rate than *C. metallidurans* CH34. Benzene transformation into catechol via dioxygenases is achieved using a single oxygen molecule, whereas monooxygenase-mediated activation converts benzene to catechol in two hydroxylation steps, with the accumulation of the toxic metabolic intermediate phenol [54]. While mercury and cadmium concentrations used in the benzene degradation assays by a resting cell approach were not equitoxic in comparison with respective MICs of *C. metallidurans* CH34 or *P. putida* F1, these concentrations corresponded to minimal heavy metal concentrations that showed observable effects on benzene degradation in both strains. Heavy metal effect on benzene catabolism in both strains could also be attributed to a general stress response, affecting other enzymatic systems that could indirectly influence the degradation rate of each compound. Nevertheless, the concentrations evaluated and their effects are in accordance with the differences within both cytoplasmic detoxification systems aforementioned in Section 4.4.

In the presence of mercury, *C. metallidurans* CH34 showed higher MICs during growth on benzene and on succinate than *P. putida* strain F1. Strain CH34 possesses a moderate mercury resistance, whereas *P. putida* F1 is a mercury-sensitive strain [35,40,41,90], which is in accordance with the effect of mercury exposure on benzene degradation assays observed. Notably, *C. metallidurans* CH34 showed a slightly higher degradation rate of benzene in the presence of mercury (32.5 µM; Figure 6A) than in the absence of this heavy metal; in contrast, *P. putida* F1 was unable to degrade benzene in the presence of mercury. The increased CH34 benzene degradation in the presence of mercury is noteworthy since the reported detrimental effect of mercury (36, 180, and 360 µM) on the catabolism of aromatic compounds in soil microbiota [14]. Although the modest increase in the benzene degradation rate was observed by strain CH34, 2-HMS production rate was lower than the unexposed condition after the 6 h, suggesting dissimilar effects for the upper or lower benzene catabolic pathway. Interestingly, the induction by arsenic of the *m*-xylene degradation genes in *P. putida* mt2 has been described [91]; therefore, the induction of some catabolic genes by some metals might not be discarded. Future studies should investigate the transcriptional regulation and possible co-activation of the benzene degradation pathway and the *mer* genes in *C. metallidurans* CH34.

Strains CH34 and F1 were able to degrade benzene in the presence of cadmium. Cd (II) (200 µM) slightly decreased the benzene degradation by *C. metallidurans* CH34 during the first 6 h (Figure 7A), but no inhibition on degradation was observed after longer incubation (Figure 7 and Appendix A). The impairment of phenol production during the short incubation time may have been caused by the destabilization of T2MO and TMO (2Fe-2S) clusters by cadmium, which has been reported for copper and zinc [92,93,94]. The partial decrease in 2-HMS production caused by cadmium could be explained by the mixed nature of *C. metallidurans* CH34 catabolic pathway, in which the cytoplasmic presence of Cd (II) may favor C12O activity instead of C23O. The location of the *catA1* gene downstream of a RpoS promoter is in accordance with an induction of its expression under cadmium stress [95,96,97]. In addition, cadmium impairs C23O activity in *Stenotrophomonas maltophilia* KB2 and *Variovorax* sp. 12S while increasing in vitro C12O activity by up to 270% [98,99]. It has been reported that cadmium binds to C23O extradiol aromatic ring cleaving dioxygenases, replacing Fe (II) in its catalytic binding site [100], but not to the C12O dioxygenase active site, which contains Fe (III) [71].

On the other hand, *P. putida* strain F1 achieved a similar benzene degradation rate in the presence or absence of cadmium, suggesting that the dioxygenase-mediated catabolism is not affected by cadmium. In accordance, the toluene dioxygenase active site containing Fe (III) showed non-susceptibility to divalent cation ligand substitution [101].

## 5. Conclusions

This study showed that *C. metallidurans* CH34 possesses a wide aromatic catabolic versatility and is able to grow on a broad range of monoaromatic compounds including BTEX compounds, chlorobenzoates, and lignin-derived aromatics. CH34 benzene degradation occurs by the activation of the aromatic ring through three BMMs and a central mixed route composed by C12O and C23O catechol dioxygenases that are differentially regulated at early and late exponential growth phases. Notably, strain CH34 degraded benzene in the presence of the toxic heavy metals mercury and cadmium. The presence of mercury (32.5 µM) slightly favored benzene degradation in *C. metallidurans* CH34; however, it inhibited benzene degradation in *P. putida* F1. Cadmium (200 µM) decreased benzene degradation by strain CH34 during short incubation (6 h) but not after longer time periods, whereas *P. putida* F1 degradation was not affected. These results indicate that *C. metallidurans* CH34 is an attractive biocatalyst for the bioremediation of BTEX in sites polluted with aromatic compounds and heavy metals such as mercury and cadmium.

## Figures and Tables

**Figure 1 microorganisms-10-00484-f001:**
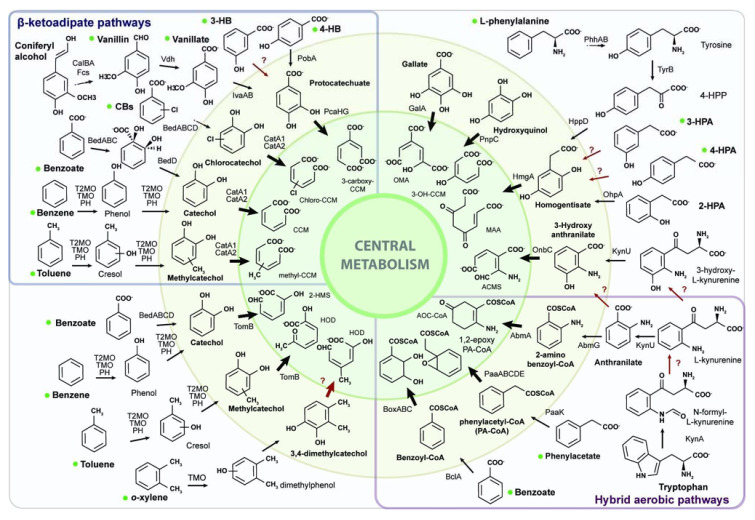
Schematic representation of aromatic peripheral and central catabolic pathways/reactions present in *C. metallidurans* CH34. The inner circle (yellow) includes ring cleavage product structures of aromatic central catabolic pathways. The outer circle (blue) includes the structure of dihydroxylated and aryl-CoA ring cleavage intermediates. Dashed arrows indicate multiple steps. Red arrows represent enzymes not encountered in *C. metallidurans* CH34 genome. Bacterial growth was observed on peripheral compounds marked with a green dot as the only carbon and energy source.

**Figure 2 microorganisms-10-00484-f002:**
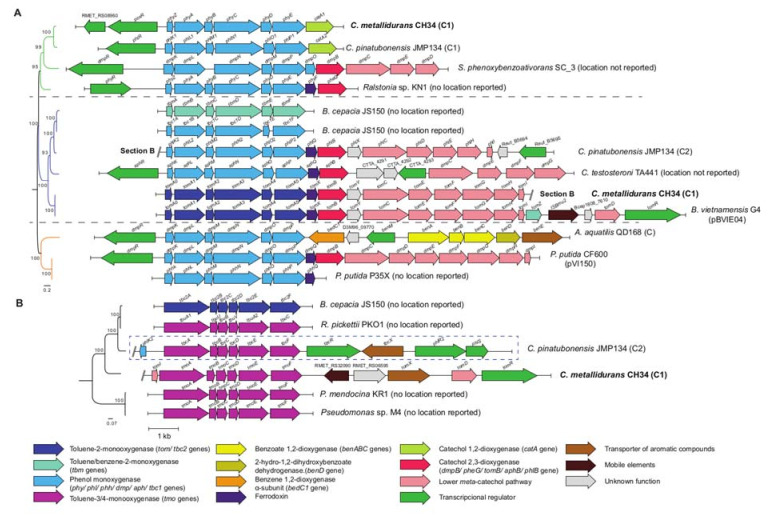
Gene cluster organization of bacterial multicomponent monooxygenases (BMMs) of *C. metallidurans* CH34 and other Proteobacteria. BMM gene clusters were classified as described by Notomista et al. (2003) into group I (**A**) and group II (**B**) according to the synteny of their hydroxylase components. Each group was further organized by phylogenetic analysis of their most divergent hydroxylase subunit (Bayesian inference method and mid-rooted). The orientations of ORFs are represented by open arrows. Genes and intergenic regions are on scale. (**A**) BMM gene cluster I. Three clades were identified by phylogenetic analysis of the α-hydroxylase subunit (PhyC, PhlN, DmpN, PhyC, TbmD, Tbc1D AphN, and TomA3). Strain CH34 *phy* and *tom* gene clusters belonged to clades 1 and 2, respectively. (**B**) BMM gene cluster II (arranged by γ-hydroxylase subunit: TmoE, TbuC, TbcF, and Tbc2F). *C. metallidurans* CH34 *tmo* gene cluster formed a singleton.

**Figure 3 microorganisms-10-00484-f003:**
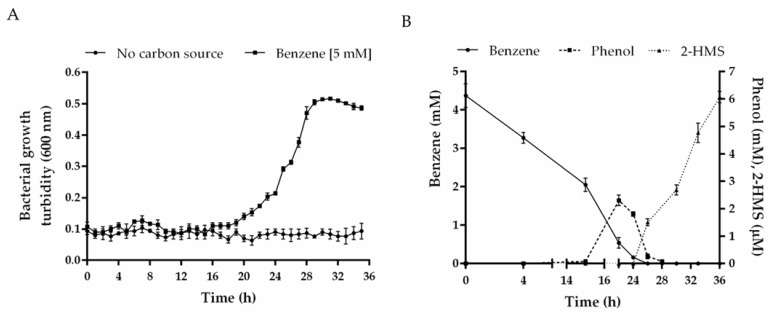
Formation of the metabolic intermediates phenol and 2-hydroxymuconic semialdehyde (2-HMS) during *C. metallidurans* CH34 growth on benzene. (**A**) CH34 cells were grown in LPTMS minimal medium using benzene (5 mM) as sole carbon and energy sources. Control assays without carbon source are also depicted. (**B**) The metabolic intermediates were analyzed by HPLC. Benzene degradation (square), phenol formation (triangle), and the generation of 2-hydroxymuconic semialdehyde (2-HMS; circle) after *meta*-cleavage of the catechol ring are indicated. Control assays without bacteria showed no degradation (data not shown). Each point is an average ± SDs of results from at least three independent assays.

**Figure 4 microorganisms-10-00484-f004:**
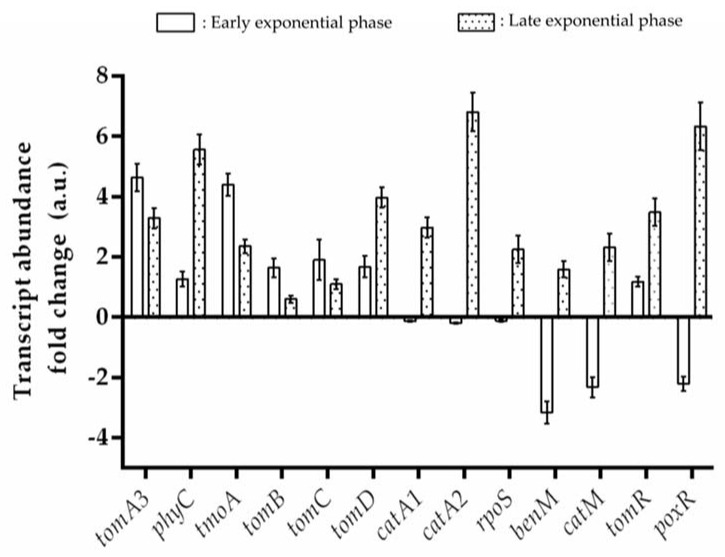
Transcriptional analysis of *C. metallidurans* CH34 benzene catabolic pathway genes. RT-qPCR assays were performed using mRNA from CH34 cells grown in LPTMS minimal medium supplemented with benzene (5 mM) until early exponential phase (turbidity at 600 nm of 0.2–0.3, ≈24 h; white column) and late exponential phase (turbidity at 600 nm of 0.5–0.6; ≈28 h, dotted column). The genes encode for toluene 2-monooxygenase (*tomA3*), toluene monooxygenase (*tmoA*), phenol hydroxylase (*phyC*), catechol 2,3-dioxygenase (*tomB*), catechol 1,2-dioxygenase (*catA1* and *catA2*), hydroxymuconic semialdehyde dehydrogenase (*tomC*), 2-hydroxymuconic semialdehyde hydrolase (*tomD*), sigma factor 38 (*rpoS*), LysR-type transcriptional regulators (*benM* and *catM*), and XylR/NtrC-type transcriptional regulators (*tomR* and *poxR*). The *gyrB* gene was used as a reference gene. The primer pairs used are listed in Appendix A. The fold-change in gene expression was calculated relative to CH34 cells grown on succinate. *p* value = 0.1%.

**Figure 5 microorganisms-10-00484-f005:**
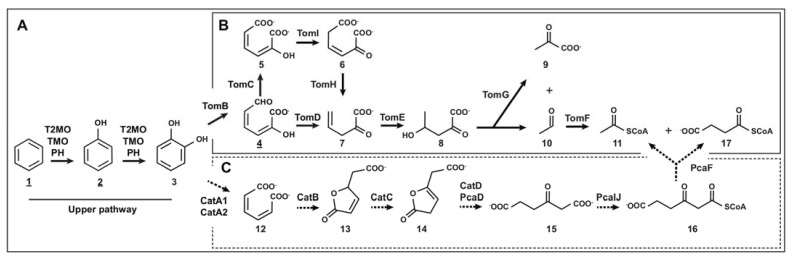
Proposed benzene catabolic pathways in *C. metallidurans* CH34. Main pathway for benzene degradation in strain CH34 is indicated with continuous arrows, whereas the oxidation pathway that was not preferred is indicated with dashed arrows. Intermediate metabolites with experimental data are underlined. (**A**) Upper pathway for benzene degradation via consecutive monooxygenation. The substrates and products are as follows: 1, benzene; 2, phenol; 3, catechol. The enzymes involved are as follows: T2MO (toluene 2-monooxygenase), TMO (toluene monooxygenase), PH (phenol hydroxylase). (**B**) Lower pathway for catechol via *meta*-cleavage. The substrates and products are as follows: 4, 2-hydroxymuconic semialdehyde; 5, 2-hydroxymuconate; 6, 4-oxalocrotonate; 7, 2-oxopenta-4-enoate; 8, 4-hydroxy-2-oxopentanoate; 9, pyruvate; 10, acetaldehyde; 11, acetyl-CoA. The enzymes involved are as follows: TomB (catechol 2,3-dioxygenase), TomC (semialdehyde-2-hydroxymuconate dehydrogenase), TomI (4-oxalocrotonate tautomerase), TomH (4-oxalocrotonate decarboxylase), TomD (semialdehyde-2-hydroxymuconate hydrolase), TomE (2-hydroxypent-2,4-dienoate hydratase), TomG (4-hydroxy-2-ketovalerate aldolase), TomF (acetaldehyde-CoA dehydrogenase). (**C**) Lower pathway for catechol via *ortho*-cleavage. The substrates and products are as follows: 12, *cis-cis* muconate; 13, muconolactone; 14, 3-oxoadipate-enol-lactone; 15, 3-oxoadipate; 16, 3-oxoadipyl-CoA; 17, succinyl-CoA. The enzymes involved are as follows: CatA (catechol 1,2-dioxygenase), CatB (muconate cycloisomerase), CatC (muconolactone delta-isomerase), CatD/PcaD (3-oxoadipate enol-lactonase), PcaIJ (3-oxoadipate CoA-transferase), PcaF (3-oxoadipyl-CoA thiolase).

**Figure 6 microorganisms-10-00484-f006:**
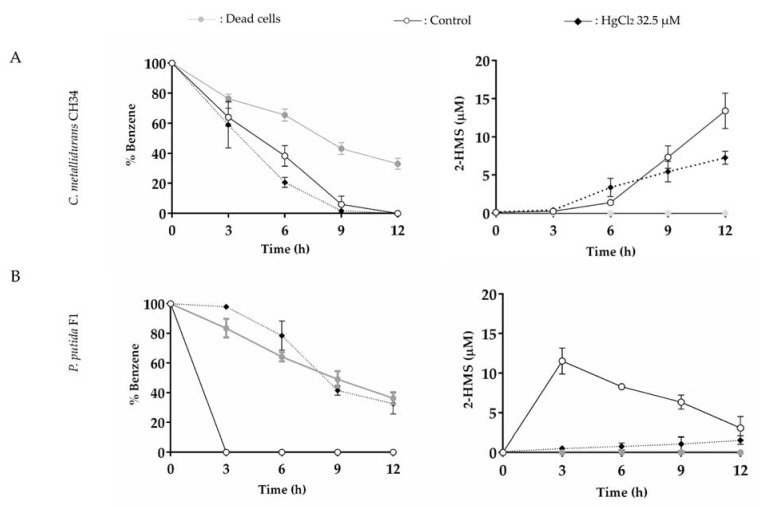
Effects of mercury on benzene degradation (5 mM) and 2-HMS formation by *C. metallidurans* CH34 and *P. putida* F1. Effects of HgCl_2_ (32.5 µM) on (**A**) *C. metallidurans* CH34 and (**B**) *P. putida* F1. Each point is an average ± SDs of results from three independent resting cell assays.

**Figure 7 microorganisms-10-00484-f007:**
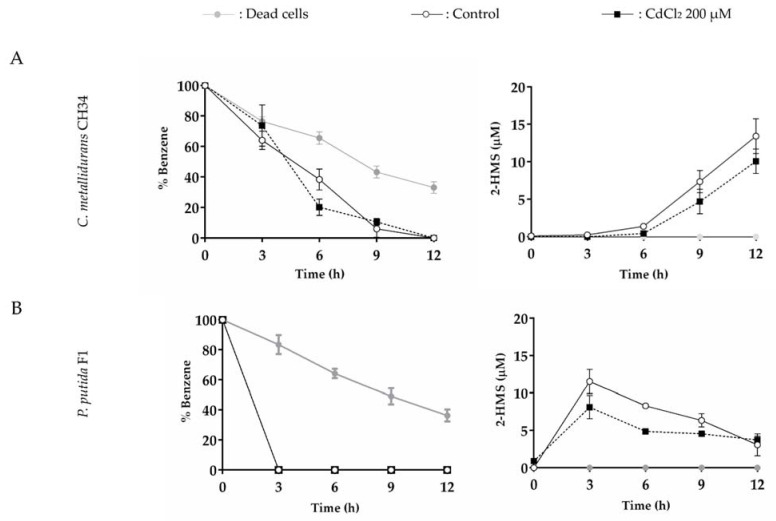
Effects of cadmium on benzene degradation (5 mM) and 2-HMS formation by *C. metallidurans* CH34 and *P. putida* F1. Effects of CdCl_2_ (200 µM) on (**A**) *C. metallidurans* CH34 and (**B**) *P. putida* F1. Each point is an average ± SDs of results from three independent resting cells assays.

**Table 1 microorganisms-10-00484-t001:** Growth of *C. metallidurans* CH34 on aromatic compounds as sole carbon and energy sources.

Carbon Source		*C. metallidurans* CH34
Monoaromatic hydrocarbons		
	Benzene	+
	Toluene ^a^	+
	Ethylbenzene	±
	*o*-Xylene	±
	*m*-Xylene	±
	*p*-Xylene	±
	*p*-Cymene	+
Polycyclic hydrocarbons		
	Phenanthrene	-
	Anthracene	-
Benzoates		
	Benzoate (BA)	++
	3-Chlorobenzoate (3-CBA)	+
	4-Chlorobenzoate (4-CBA)	-
	3,5-Dichlorobenzoate (3,5-CBA)	±
	2-Hydroxybenzoate (salicylate)	+++
	3-Hydroxybenzoate (3-HBA)	+++
	4-Hydroxybenzoate (4-HBA)	+++
	2-Aminobenzoate (anthranilate)	-
	4-Aminobenzoate (*p*ABA)	++
	4-Isopropylbenzoate (*p*-cumate)	-
Phenylacetates		
	Phenylacetate (PA)	+
	3-Hydroxyphenylacetate (3-HPA)	+++
	4-Hydroxyphenylacetate (4-HPA)	++
	Cinnamate	++
	Vanillin	+++
	Vanillate	+ ^b^
Aromatic amino acids		
	*L*-Phenylalanine	++
	*L*-Tyrosine	+ ^b^
Other aromatic compounds		
	Benzamide	-
	Nitrobenzene	-
	*m*-Toluate	-
	Hydroxyquinol (HQ)	-
	Gallate	-
Other non-aromatic compounds		
	Succinate	+

-, No growth; ±, bacterial aggregation; +, turbidity_600nm_ > 0.10; ++, turbidity_600nm_ > 0.25; +++, turbidity_600nm_ > 0.50. Turbidity at 600nm was assessed every 24 h for a period of 7 days. ^a^ Compound provided in gaseous phase. ^b^ David et al. (1996).

**Table 2 microorganisms-10-00484-t002:** Minimal inhibitory concentrations of cadmium and mercury in *C. metallidurans* CH34 and *P. putida* F1 using succinate or benzene as sole carbon and energy source.

	Mercury (µM)			Cadmium (mM)		
Strain	Succinate	Benzene	Fold Change	Succinate	Benzene	Fold Change
*C. metallidurans* CH34	12.5	2	6.25	4	0.8	5
*P. putida* F1	3.25	1	3.25	4	0.4	10

Data were collected after an incubation of 48 h. Values were determined in assays in triplicate.

## Data Availability

Not applicable.

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
