# Peer review of "Cupriavidus metallidurans* CH34 Possesses Aromatic Catabolic Versatility and Degrades Benzene in the Presence of Mercury and Cadmium"

_microorganisms, 2022, doi:10.3390/microorganisms10020484_

Round 1
Reviewer 1 Report
We are faced with a nice piece of work in which authors correlate the aromatic compounds biodegradative ability of a strain of Cupriavidus metallidurans with the inherent resistance of this strain towards cadmium and mercury. The approach is based on the hypothesis of an effect of the presence of these heavy metals to the catabolic ability of the strain to degrade benzene.
As a suggestion, I would appreciate if in Supplementary table 2, primer efficiencies were indicated.
My feeling is that the work is well argued and the results have been acquired in an appropriate technical approach.
However, my main concern after reading the entire manuscript is that there is a point that, under my opinion, should be discussed further: authors report a MIC when the strain under study and a control strain (Pseudomonas putida F1) are cultured in minimal media containing 5 mM benzene as carbon source and mercury (MIC=2 μM for C. metallidurans and 1 μM for P. putida) or cadmium (MIC=0.8 μM for C. metallidurans and 0.4 μM for P. putida); my understanding is that these heavy metal concentrations (or higher) inhibiting the growth of these strains, affect to different systems in the metabolic network of the microorganisms. However, when authors report the effect of these heavy metals specifically over benzene metabolism using a resting cell approach, they should increase the concentrations to 32.5 μM for mercury and 200 μM for Cd. Therefore, my point is that these differences in working concentrations when looking at benzene catabolism and the MICs observed should be further detailed in the discussion section. It is evident that the toxicity of these heavy metals at MIC concentrations affect more critical metabolic systems than the benzene catabolism, which is effectively affected at substantially higher concentrations. I suggest that authors include a commentary about this point in the manuscript.
As minor points:
Line 76. I suggest to restrict the references [25, 35, 37-41] to [39,40], or as much, including also 35 and 41.
Lines 143-144. Authors include a reference as (Duran et al., 2019). Please, use the indications for references according to the standards of the journal. I suppose that this is reference [64]. Please if it is the case, renumber the list of references.
From lines 248 to 467. Please, use italics for scientific names of the strains
From lines 254 to 395. Please, use italics for the names of the genes.
As a comment on lines 257 to 262. Considering exclusivelly the metabolism of phenylacetate, I could not consider phenylacetyl-CoA ligase as a peripheral reaction. Styrene, ethylbenzene and other compounds converge at phenylacetate level. Only if you consider metabolism of phenylalkanoates with a side chain with an even number of carbon atoms, the convergence point is phenylacetyl-CoA. Thus, taking into account that we do not know if this strain is capable of metabolizing these phenylalkanoates I cannot correct or maintain PaaK as a peripheral or central activity. I will accept the decision of the authors.
From lines 326 to line 341. Prefix indication of ortho, meta and para in the molecules, i.e. m-toluate should be written in italics. Same for 4-aminobenzoate (pABA) in table 1 and other positions.
From line 319 to 578. I suggest to use italics for ortho- and meta-cleavage, because indicate specific positions in the molecules.
In table 1. Authors indicate the growth using L-Phenylalanine but they do not indicate if used Tyrosine is a L-, D, or LD-amino acid.
Lines 354-355. Authors indicate the disappearance of putative 2-HMS after 48 h of growth culture, in Figure 3B (not 3A as indicated in the text) we cannot see this effect. In the figure we only can see the accumulation of the compound at 36 hours as a by-product, we cannot see the decrease of this compound.
Figure 3. Comparing panels A and B, at 24 hours of incubation, benzene has disappeared from the medium, although the strain is only in the early logarithmic phase of growth. Can the authors explain the growth reached in the 6-8 hours later to benzene catabolism?
Lines 440 to 449. Although in Materials and Methods section authors detailed the condition of the culture, please, to increase the interpretation of the data by readers, summarize the benzene concentration used, the amount of cells used in the experiments and other important parameters of the experiment.
Figures 6 and 7. Although I understand that results showed in figure 3 correspond to the growing of the strain in a minimal medium and figures 6 and 7 show resting cells experiments, which can be exemplified by the disappearance in short time of the benzene used in the experiment using resting cells, my feeling is that authors should make some commentary about the appearance of 2-HMS: in figure 3 this compound starts to be detected when benzene is exhausted from the media. However, in figure 6 this compound appears very early. My first supposition analyzing figure 3B was that 2-HMS really appears from the metabolism of phenol that appears in the media in base to the concordance between phenol consumption and 2-HMS presence. Against that proposal was the concentration of phenol and 2-HMS observed. Taking this into account I miss to see (i) behaviour of the wild type strain using directly phenol as carbon source to complete figure 3 and, (ii) a panel in figures 6 and 7 showing phenol present in the media from resting cells experiments.
Author Response
Comments and suggestions:
“We are faced with a nice piece of work in which authors correlate the aromatic compounds biodegradative ability of a strain of Cupriavidus metallidurans with the inherent resistance of this strain towards cadmium and mercury. The approach is based on the hypothesis of an effect of the presence of these heavy metals to the catabolic ability of the strain to degrade benzene.”
Dear reviewer, we kindly appreciate the thorough revision of our manuscript and the helpful comments. Several modifications have been performed in order to improve the manuscript which are highlighted in yellow.
Comment No. 1:
“As a suggestion, I would appreciate if in Supplementary table 2, primer efficiencies were indicated.”
A: As suggested by the reviewer, the primer efficiencies were added to the Supplementary Table 2.
Comment No. 2:
“My feeling is that the work is well argued and the results have been acquired in an appropriate technical approach.
However, my main concern after reading the entire manuscript is that there is a point that, under my opinion, should be discussed further: authors report a MIC when the strain under study and a control strain (Pseudomonas putida F1) are cultured in minimal media containing 5 mM benzene as carbon source and mercury (MIC=2 μM for C. metallidurans and 1 μM for P. putida) or cadmium (MIC=0.8 μM for C. metallidurans and 0.4 μM for P. putida); my understanding is that these heavy metal concentrations (or higher) inhibiting the growth of these strains, affect to different systems in the metabolic network of the microorganisms. However, when authors report the effect of these heavy metals specifically over benzene metabolism using a resting cell approach, they should increase the concentrations to 32.5 μM for mercury and 200 μM for Cd. Therefore, my point is that these differences in working concentrations when looking at benzene catabolism and the MICs observed should be further detailed in the discussion section. It is evident that the toxicity of these heavy metals at MIC concentrations affect more critical metabolic systems than the benzene catabolism, which is effectively affected at substantially higher concentrations. I suggest that authors include a commentary about this point in the manuscript.”
A: Thank you for the critical revision of our manuscript, especially for pointing out the different heavy metal concentrations used in the resting cell experiment in comparison to the observed MICs. The resting cell assay allows us to assess the effect of the heavy metals on the previously induced enzymatic machinery under non-growing conditions. While mercury and cadmium concentrations used in benzene degradation assays were not equitoxic in comparison with respective MICs of C. metallidurans CH34 or P. putida F1, these concentrations corresponded to the minimal heavy metal concentrations that showed observable effects in the benzene resting cell assays on both strains. We understand that other enzymatic systems, not only the benzene catabolism, may be involved and could indirectly affect the degradation rate of each compound. Nevertheless, the concentrations evaluated, and its effects are in accordance with the differences within both cytoplasmic detoxification systems aforementioned in section 4.4. Some of this discussion is now explicitly stated in section 4.5.
Comment No. 3:
“Line 76. I suggest to restrict the references [25, 35, 37-41] to [39,40], or as much, including also 35 and 41.”
A: As suggested by the reviewer the references were reduced to [35, 39-41], (now 35, 38-40].
Comment No. 4:
“Lines 143-144. Authors include a reference as (Duran et al., 2019). Please, use the indications for references according to the standards of the journal. I suppose that this is reference [64]. Please if it is the case, renumber the list of references.”
A: Thank you for pointing out that mistake, Durán et al., 2019 was the reference [64], now [50]. After that we have rearranged the reference numeration. This was modified in the manuscript.
Comment No. 5:
“From lines 248 to 467. Please, use italics for scientific names of the strains
From lines 254 to 395. Please, use italics for the names of the genes. “
A: Thank you for letting us know about this issue. Due to some problems with the automatic conversion of the submission page, the italics are not present in the result section of the pdf document, but they are in the MS Word document (microorganisms-1587111.docx). For this reason, we are also submitting a pdf document with the revision highlighted in yellow.
Comment No. 6:
“As a comment on lines 257 to 262. Considering exclusivelly the metabolism of phenylacetate, I could not consider phenylacetyl-CoA ligase as a peripheral reaction.
Styrene, ethylbenzene and other compounds converge at phenylacetate level. Only if you consider metabolism of phenylalkanoates with a side chain with an even number of carbon atoms, the convergence point is phenylacetyl-CoA. Thus, taking into account that we do not know if this strain is capable of metabolizing these phenylalkanoates I cannot correct or maintain PaaK as a peripheral or central activity. I will accept the decision of the authors.”
A: We understand your thoughts regarding this topic. For the same reasons as you stated, we will rather prefer to keep the classification of this activity as a peripheral reaction.
Comment No. 7:
“From lines 326 to line 341. Prefix indication of ortho, meta and para in the molecules, i.e. m-toluate should be written in italics. Same for 4-aminobenzoate (pABA) in table 1 and other positions.”
A: Thank you for pointing out this issue. As stated previously is the same issue as species names and genes only observed in the result section of the pdf document.
Comment No. 8:
“From line 319 to 578. I suggest to use italics for ortho- and meta-cleavage, because indicate specific positions in the molecules.”
A: As suggested by the reviewer we have added italics to ortho- and meta- prefixes.
Comment No. 9:
“In table 1. Authors indicate the growth using L-Phenylalanine but they do not indicate if used Tyrosine is a L-, D, or LD-amino acid.”
A: The growth on tyrosine of strain CH34 was performed on L-Tyrosine. This was not performed by our study, but referenced from David et al., 1996, reference [47]. As suggested, we added which enantiomer was used by David et al., 1996.
Comment No. 10:
“Lines 354-355. Authors indicate the disappearance of putative 2-HMS after 48 h of growth culture, in Figure 3B (not 3A as indicated in the text) we cannot see this effect. In the figure we only can see the accumulation of the compound at 36 hours as a by-product, we cannot see the decrease of this compound.”
A: That is correct. Figure 3A does not show the production after 36 h. In consequence, the phrase “which disappeared after 48 h (Figure 3A)” was removed.
Comment No. 11:
“Figure 3. Comparing panels A and B, at 24 hours of incubation, benzene has disappeared from the medium, although the strain is only in the early logarithmic phase of growth. Can the authors explain the growth reached in the 6-8 hours later to benzene catabolism?”
A: As stated, growth of strain CH34 on benzene only reached the early exponential phase at 24 h. Nevertheless, concentration of phenol and 2-HMS can still be measured suggesting these intermediary compounds could support C. metallidurans CH34 growth after benzene depletion.
Comment No. 12:
“Figures 6 and 7. Although I understand that results showed in figure 3 correspond to the growing of the strain in a minimal medium and figures 6 and 7 show resting cells experiments, which can be exemplified by the disappearance in short time of the benzene used in the experiment using resting cells, my feeling is that authors should make some commentary about the appearance of 2-HMS: in figure 3 this compound starts to be detected when benzene is exhausted from the media. However, in figure 6 this compound appears very early. My first supposition analyzing figure 3B was that 2-HMS really appears from the metabolism of phenol that appears in the media in base to the concordance between phenol consumption and 2-HMS presence. Against that proposal was the concentration of phenol and 2-HMS observed. Taking this into account I miss to see (i) behaviour of the wild type strain using directly phenol as carbon source to complete figure 3 and, (ii) a panel in figures 6 and 7 showing phenol present in the media from resting cells experiments.”
A: We understand your concerns regarding the phenol and 2-HMS production by strain CH34. As stated, the data suggests that 2-HMS production appears after phenol consumption on growth experiments on benzene by strain CH34 (Figure 3B). For the resting cell assay, even though this is not as evident as the growth experiments, phenol is detected at 3 h (phenol was determined in resting cell experiments for strain CH34, Figure S1), while 2-HMS is detected at 6 h (Figure 6 & 7). Due to the nature of the resting cell assay, the appearance of phenol or 2-HMS does not necessarily correspond with the complete depletion of benzene and phenol, respectively, which could be accounted for by the enzymes currently induced at the starting point of the resting cell experiment. The data obtained by the resting cell assays by strain CH34 supports that phenol is the first intermediate, while 2-HMS presence appears after phenol production. Moreover, this suggests that benzene catabolism by strain CH34 is intricately regulated on growing cells, probably requiring the complete depletion of some intermediates to continue to the next enzymatic step. Some part of this response was incorporated in the main manuscript (Result section 3.6 and Figure S1 caption).
Reviewer 2 Report
The reviewed manuscript presents a well-conducted, rigorous and complete study on the biodegradation potential of the strain Cupriavidus metallidurans CH34, known as a multi heavy metal resistant bacterial model, towards various aromatic compounds. Focus is on the benzene catabolism pathways (from a metabolite and gene expression points of view) and on the impact of two metals (mercury and cadmium) on its biodegradation kinetics. The discussion is convincing with a substantial bibliography.
This very interesting paper is publishable in Microorganisms after detailing some points and correcting the minor errors listed below.
Introduction
- L55: The reference 19 is listed after the reference 20-22 (L53) in the text.
- L62: Is the species of strain Cupriavidus JMP134 not necator instead of pinatubonensis?
Materials and Methods
- L109: Remove the references with authors (“Mergeay; Rojas”).
- L196-204: The protocol seems to be repeated twice and is difficult to follow. Make it clearer.
- L215-217: Here again the “glass tubes capped with … caps” is repeated with no reason. Modify.
- L232: Specify the method used to kill the cells.
- L237: Specify the granulometry of the HPLC column, the injection volume and the wavelength(s) used.
Results
- In this section, no name of the bacteria cited is written in italics except in the Figure captions. Modify.
- Table 1: Specify in the caption or in the footnote, the time after which the turbidity is measured to class the aromatic compounds as a source of bacterial growth or not (7 days as stated in the M&M?).
- L356: Write the full name of the molecule for 2-HMS and remind Figure 1 maybe.
- L399-400: Specify the growth time or growth time range for the early exponential phase and the late exponential phase.
- Figure 6 caption as well as in the text: Specify the initial concentration of benzene.
Discussion
- L488: Remove one of the two “of” in the term “the harboring of of C1 and C2 genes”
- Section 4.3. /Discussion of Figure 3: Could the authors comment more specifically the fact that benzene is almost completely degraded when the bacterial growth starts to be observed (around 20 h of incubation)?
Author Response
Comments and Suggestions for Authors
The reviewed manuscript presents a well-conducted, rigorous and complete study on the biodegradation potential of the strain Cupriavidus metallidurans CH34, known as a multi heavy metal resistant bacterial model, towards various aromatic compounds. Focus is on the benzene catabolism pathways (from a metabolite and gene expression points of view) and on the impact of two metals (mercury and cadmium) on its biodegradation kinetics. The discussion is convincing with a substantial bibliography.
This very interesting paper is publishable in Microorganisms after detailing some points and correcting the minor errors listed below.
Dear reviewer, We thank you for the time dedicated to the revision, the positive feedback, and the comments to our study. After the revision we have improved the quality of the manuscript.
Comment No. 1:
“Introduction
L55: The reference 19 is listed after the reference 20-22 (L53) in the text.”
A: Thanks for noticing this reference. The mentioned references were rearranged, and every other reference was double-checked.
Comment No. 2:
“L62: Is the species of strain Cupriavidus JMP134 not necator instead of pinatubonensis?”
A: Cupriavidus pinatubonensis JMP134, formerly Alcaligenes eutrophus, Ralstonia eutropha, Wautersia eutropha, and Cupriavidus necator, was reclassified by Sato et al., 2006 during the species description of C. pinatubonensis sp. nov. (Proposal to reclassify Ralstonia sp. JMP134 to C. pinatubonensis JMP134)
Comment No. 3:
“L109: Remove the references with authors (“Mergeay; Rojas”)”
A: Thank you for noticing this mistake. The references were removed.
Comment No. 4:
“L196-204: The protocol seems to be repeated twice and is difficult to follow. Make it clearer.
L215-217: Here again the “glass tubes capped with … caps” is repeated with no reason. Modify.” A: Thank you for your feedback, we have performed some changes to the Material and Methods section to make it simpler.
Comment No. 5:
“L232: Specify the method used to kill the cells.”
A: We have added the method used to kill the cells
Comment No. 6:
“L237: Specify the granulometry of the HPLC column, the injection volume and the wavelength(s) used.”
A: The requested information was provided.
Comment No. 7:
“In this section, no name of the bacteria cited is written in italics except in the Figure captions. Modify.“
A: Thank you for letting us know about this issue. Due to some problems with the automatic conversion of the submission page, the italics are not present in the result section of the pdf document, but they are in the MS Word document (microorganisms-1587111.docx). For this reason, we are also submitting a pdf document with the revision highlighted in yellow.
Comment No. 8:
“Table 1: Specify in the caption or in the footnote, the time after which the turbidity is measured to class the aromatic compounds as a source of bacterial growth or not (7 days as stated in the M&M?).”
A: As suggested by the reviewer this was incorporated in the caption of the table. Turbidity at 600nm was assessed every 24 h for a period of 7 days to determine growth on any of the aromatic compounds tested.
Comment No. 9:
“L356: Write the full name of the molecule for 2-HMS and remind Figure 1 maybe.”
A: As suggested, this was performed in the manuscript (L356).
Comment No. 10:
“L399-400: Specify the growth time or growth time range for the early exponential phase and the late exponential phase.”
A: The early exponential phase was defined at turbidity600nm= 0.2-0.3 (~24 h), while the late exponential phase was defined at turbidy600nm= 0.5-0.6 (~28 h). This information is now stated in the caption of Figure 4.
Comment No. 11:
“Figure 6 caption as well as in the text: Specify the initial concentration of benzene.”
A: As suggested, this information was added in the caption of Figures 6 and 7, and in the text.
Comment No. 13:
“L488: Remove one of the two “of” in the term “the harboring of of C1 and C2 genes””
A: This was modified in the main manuscript.
Comment No. 14:
“Section 4.3. /Discussion of Figure 3: Could the authors comment more specifically the fact that benzene is almost completely degraded when the bacterial growth starts to be observed (around 20 h of incubation)?”
A: As stated, growth of strain CH34 on benzene only reached the early exponential phase at 24 h. Nevertheless, concentration of phenol and 2-HMS can still be measured suggesting these intermediary compounds could support CH34 growth after benzene depletion.